# Underlying Role of Rumination-Mediated Attachment Style Plays in PTSD after TIA and Stroke

**DOI:** 10.3390/brainsci12091118

**Published:** 2022-08-23

**Authors:** Kaiping Zhou, Linjing Zhang, Tonggui Li, Weiping Wang

**Affiliations:** 1Key Laboratory of Neurology of Hebei Province, Department of Neurology, The Second Hospital of Hebei Medical University, Shijiazhuang 050051, China; 2Department of Neurology, Peking University Third Hospital, Beijing 100191, China; 3School of Psychological and Cognitive Sciences, Peking University, Beijing 100191, China; 4Beijing Key Laboratory of Behavior and Mental Health, Peking University, Beijing 100191, China

**Keywords:** stroke, transient ischemic attack, post-traumatic stress disorder, attachment, rumination, medication adherence

## Abstract

Objective: Attachment and rumination were examined as the intermediary variables on post-traumatic stress disorder and medication compliance in stroke or TIA patients. Methods: A total of 300 participants with stroke or TIA from the Second Hospital of Hebei Province were selected. Patients accomplished NIHSS, ABCD^2^, ECR, RSQ, and RRS on admission. After 3 months, the PCL-C and MMAS were collected. Results: In the stroke or TIA patients, the incident of PTSD was 7.7%; PTSD scores were significantly associated with attachment anxiety (r = 0.225, *p* < 0.01), symptom rumination (r = 0.197, *p* < 0.01), and obsessive thinking (r = 0.187, *p* < 0.01). After the Sobel test analysis and verification by the Baron and Kenny’s stepwise approach we found that ruminant mediated the relationship between attachment anxiety and PTSD; obsessive thinking mediated the relationship between attachment anxiety and PTSD. Conclusions: The relationship between attachment anxiety and PTSD was positively predicted by rumination and obsessive thinking. Adult attachment style, rumination, and PTSD scores may not predict medication compliance.

## 1. Background

Transient ischemic attack (TIA) and ischemic stroke are two of the leading causes of death and disability and presents a worldwide health care problem. Although post-traumatic stress disorder (PTSD) commonly occurs due to traumas caused by exposure to natural disasters or violence, [1,2], PTSD has also been investigated in patients that suffered from life-threatening medical conditions including ischemic cerebrovascular events, and the prevalence and correlates of PTSD after TIA or ischemic stroke have been investigated in previous studies [3,4,5,6,7,8]. The crude prevalence of PTSD after TIA or ischemic stroke was varied from 12.9–31% [3,4,5,6,7,8]. There are relatively few studies on PTSD after stroke, with conflicting results for many predictors of PTSD after stroke. By comparing the general demographic characteristics (age, sex, marital status, education level, ethnicity, etc.) and the impact degree of stroke between the post-traumatic stress disorder population and the non-post-traumatic stress disorder population, no consistent conclusion has been reached. This may be related to differences in traumatic experience between medical and non-medical PTSD [9,10,11]. Participants with PTSD were more likely to have low compliance to medications than other participants [4].

Apart from the lack of knowledge about the risk factors for PTSD after TIA and stroke, psychological mechanisms that can prevent PTSD development have not been investigated in TIA and stroke patients until now. There has been increasing research showing a link between attachment and PTSD [12,13,14,15,16,17,18]. A study has begun to suggest that attachment insecurity is associated with the severity of symptoms of PTSD [19]. Insecure attachment was also found to mediate PTSD in abused children, [17] inflammatory bowel disease, [13] and HIV patients [15]. Attachment style in therapy may improve the quality of life and prognosis of the patients [15,20]. Rumination has also been shown to be associated with traumatic stress and may be a risk factor for PTSD [20,21,22,23]. Repeated rumination has been linked to PTSD in children after earthquakes, in first time postpartum women, and in employees who have experienced occupational accidents as well as firefighters [24,25,26,27,28,29].

Such research has shown that attachment and rumination were strongly and predictively associated with PTSD. Researchers have further discovered that rumination is a mediator of attachment to physical health and emotional regulation [30,31,32,33]. However, these associations were estimated in a general population, and whether this association still exists in patients suffering from TIA or stroke was ambiguous. 

This study intends to explore the relationship between attachment, rumination, and PTSD in stroke patients as well as the impact on drug compliance, in order to provide some reference for early psychological nursing measures.

## 2. Methods

This study was a prospective longitudinal observational study of acute stroke and TIA patients. Three hundred participants were recruited at the Department of Neurology in The Second Hospital of Hebei Medical University, in the period from December 2018 to September 2019. This study was approved by the Research Ethics Committee of the Second Hospital of Hebei Medical University. The procedure of the study is described in Figure 1. Patients were eligible if they were between 18 and 80 years of age and had acute stroke and TIA (all were first time, NIHSS total score ≤ 15, ABCD total score > 1) and those without serious complications such as heart failure and gastrointestinal bleeding. Key exclusion criteria were a history of any mental illness or other traumatic events (e.g., traffic accident, other major illness, bereavement) in the last 6 months or recurrence of stroke or TIA within 3 months or those who were unable to complete the scale due to aphasia or severe cognitive impairment. Patients and their families were given an oral and written explanation of the study and signed informed consent. 

Eligible patients were assessed with either a National Institute of Health Stroke Scale (NIHSS), which assessed the patient for functional impairment, or an ABCD^2^ score, which predicted stroke risk in TIA patients. Once included, patients were evaluated for attachment archetypes by the Relationships Scales Questionnaire (RSQ), attachment dimensions were measured by the Experiences in Close Relationships Inventory (ECR), and the Ruminative Responses Scale (RRS), which assessed the rumination dimensions were also measured [34,35]. The PTSD symptoms and medicine compliance were measured 3 months after stroke. The grade of PTSD symptomatology was assessed using a civilian version of PCL (PCL-C). It contains 17 items and screens in a variety of clinical settings [34]. The Morisky Medication Adherence Scale (MMAS) is a 8-item scale, which is a simple and practical tool for evaluating medication compliance for discharged patients [36]. 

In November 1994, according to DSM-W, the PTSD Checklist-Civilian Version (PCL-C) was created by the United States. Seventeen items are graded into five levels (not at all, a little, moderately, quite a lot, very much). It is a multi-dimensional observation tool for PTSD including re-experiencing, avoidance, numbing, and hyperarousal [37]. The higher the cumulative score (17 to 85), the greater the likelihood of PTSD. A score of 17 to 37 indicates no significant PTSD symptoms, 38 to 49 means some degree of PTSD symptoms, and 50 to 85 shows a high level of PTSD symptoms. With respect to the MMAS, the first seven items are yes/no and the last is a 5-point rating [38]. The scoring criteria are low adherence (<6), medium adherence (6–8), and high adherence (8) [36]. Moreover, the patients’ age, sex, education, and occupation were also recorded.

Statistics were processed using SPSS version 20 (SPSS, Inc., Chicago, IL, USA). ANOVA was used to observe the differences in various indicators among subjects with different attachment types, PTSD degrees, and medication compliance. To test the mediational model, both the Sobel test and bootstrapping procedure were adopted. 

## 3. Results

Among the 300 patients, 20 patients had cerebral infarction again, two were lost to follow-up, one had fracture surgery, one had cerebrovascular bypass surgery, and two had an incomplete questionnaire due to memory decline after stroke during 3 month follow-up. Therefore, 274 were effective, comprising 185 (67.9%) males (aged mean ± SD, 54.78 ± 10.58) and 88 (32.1%) females (aged mean ± SD, 58.95 ± 9.30). A total of 60% of the patients had an education level above primary school. A total of 253 patients had ischemic stroke, nine patients had hemorrhagic stroke, and 12 patients with TIA were included in the study.

The NIHSS score was the mean ± SD, 2.18 ± 2.47 and the ABCD^2^ score was the mean ± SD, 2.58 ± 1.08). A total of 7.7% (21/274) of patients had significant PTSD. In addition, the low medication adherence was 149 (54.4%), the medium adherence was 118 (43.1%), the high adherence was 7 (2.5%). Sociodemographic information of the participants are summarized in Appendix A.

### 3.1. Intercorrelations between Study Variables

The PTSD scores were significantly associated with attachment anxiety (r = 0.225, *p* < 0.01), symptom rumination (r = 0.197, *p* < 0.01), and obsessive thinking (r = 0.187, *p* < 0.01). In addition, attachment anxiety was significantly associated with symptom rumination (r = 0.44, *p* < 0.01) and obsessive thinking (r = 0.35, *p* < 0.01). As for medication compliance, none of the measures were significantly correlated with it (Table 1).

### 3.2. Mediation Analysis

To test the mediating effect, we employed the stepwise test regression coefficient method (Sobel test). In Model 1, attachment anxiety had a positive predictive effect on the PTSD scores (β = 0.225, *p* < 0.01) (i.e., the higher the degree of attachment anxiety, the higher the PTSD score). In addition, attachment anxiety had a significant positive predictive effect on the mediating factor (symptom rumination) (Model 2, β = 0.440, *p* < 0.01). From Model 3, it was found that the positive predictive effect of attachment anxiety on the PTSD scores decreased from β = 0.225 (*p* < 0.01) to β = 0.171 (*p* < 0.05) after the addition of symptomatic ruminant (Table 2). According to the method of Baron and Kenny (1986) [39], symptomatic ruminant mediates the relationship between attachment anxiety and PTSD (Figure 2).

In Model 4 and Model 5, attachment anxiety had a positive predictive effect on the PTSD scores (β = 0.225, *p* < 0.01) and the mediating factor (obsessive thinking) (Model 5, β = 0.281, *p* < 0.01). From Model 6, it was found that the positive predictive effect of attachment anxiety on the PTSD scores decreased from β = 0.225 (*p* < 0.01) to β = 0.215 (*p* < 0.05) after the addition of obsessive thinking (Table 3). According to the method by Baron and Kenny (1986) [39], obsessive thinking mediates the relationship between attachment anxiety and PTSD (Figure 3).

Bootstrap was used to test the mediating effect (Wen Zhonglin and Ye Baojuan, 2014) [40]. The estimated mediating effect of symptom rumination between attachment anxiety and PTSD scores was 23.66% with a 95% confidence interval [0.279,3.074], indicating a significant mediating effect. In the meantime, the estimated mediating effect of the obsessive thinking between attachment anxiety and PTSD scores was 4.12%, with a 95% confidence interval [0.786,3.091], indicating a significant mediating effect.

## 4. Discussion

This study contributes to verification between attachment, rumination, and PTSD and, more explicitly, to examine the mediation where in the different dimensions of attachment could predict PTSD in stroke or TIA patients directly and indirectly through the role of specific ruminant regulatory processes. Furthermore, there have been no studies using mediating and moderating models of these variables in patients with stroke or TIA. Our findings provide areas for intervention in the clinical practice of stroke or TIA patients with PTSD.

As noted earlier, the results indicated that symptom rumination and obsessive thinking mediated the relationship between attachment anxiety and PTSD. In other words, stroke or TIA patients who tended to attachment anxiety also suffer more from PTSD, which was connected to symptom rumination and obsessive thinking. The occurrence of stroke and TIA caused more ineffective and problematic thinking among patients with higher attachment anxiety. In addition, anxious people tend to over activate their emotions and pain, and are more likely to be ruminating [41]. Sudden neurological dysfunction is a physical and psychological trauma that may activate the patient’s attachment system. Differences in attachment styles affect how patients solve and express problems. Secure attachment can increase resilience, and patients will take the initiative to talk to others and seek help to protect themselves from the negative effects of trauma. However, patients with attachment anxiety repeatedly think about the experience of stroke or the impact of sequelae of stroke on life, which brings physical and mental discomfort to patients. Lias Bishop et al. suggested that rumination caused PTSD, and the underlying mechanism may be experiential avoidance [23]. In conclusion, these results suggest that high attachment anxiety may be a risk factor for PTSD. When stroke and TIA occur suddenly, patients with attachment anxiety seem to experience symptom rumination and obsessive thinking, trying to cope with illness in an unhealthy way.

The results of this study indicated that neither adult attachment style nor rumination had a predictive effect on medication compliance, which was not consistent with other studies. It may be that medication compliance is more likely to be related to the patient’s age, underlying disease status, financial status, social support, disease level, disease awareness, and other factors. 

This study has clear clinical implications. Since this study discovers that attachment style and rumination play a role in how individuals cope with PTSD following stroke or TIA, clinicians should focus on the patients’ attachment style and rumination and use them to guide treatment. Clinicians can provide patients with specific psychological education to reduce ruminative thoughts. Clinicians can provide patients with specific psychological education to reduce their ruminative thoughts. For example, medical staff should improve education work to make patients correctly understand the occurrence and prognosis of stroke. Second, mindful attention is an important modulator of the relationship between rumination and all aspects of PTSD symptoms. Therefore, mindfulness meditation is a promising basis for interventions. In recent years, there have been many efforts to prevent PTSD in the general population [42,43]. However, the prevention of PTSD after stroke is rare and there is no expert consensus on it. We still need to explore it in large samples.

We acknowledge several limitations inherent in this study. First, there is the fact that attachment styles vary with age. Then, the subjects included were mild and moderate stroke patients, while the patients with severe stroke were unable to complete the questionnaire. Our cohort was represented by patients with mild to moderate stroke, so was only a small proportion of patients with apparent PTSD. Findings require cautious interpretation. The correlations between the PTSD scores and the other variables is thus probably mostly influenced by the scores of the large number of patients with mild PTSD. This makes it difficult to make inferences about the small number of patients with significant PTSD symptoms. Therefore, the results of the study may not be generalizable to patients with severe stroke. Our results deserve further evaluation in a larger prospective cohort study. Additionally, the title of the adult attachment scale used in this study was translated from the English version, which is indeed difficult for elderly patients with stroke and TIA to understand. Meanwhile, there was no further search for the reason that each indicator was not correlated with drug compliance. These variations must be considered when interpreting the results of studies of PTSD after stroke.

## 5. Conclusions

In the stroke or TIA patients, the incidents of PTSD was 7.7%; Our study supports the underlying role of that symptom rumination and obsessive thinking mediated attachment anxiety plays in PTSD after TIA and stroke. Therefore, attention should be paid to the mental state of stroke patients in the early stage.

## Figures and Tables

**Figure 1 brainsci-12-01118-f001:**
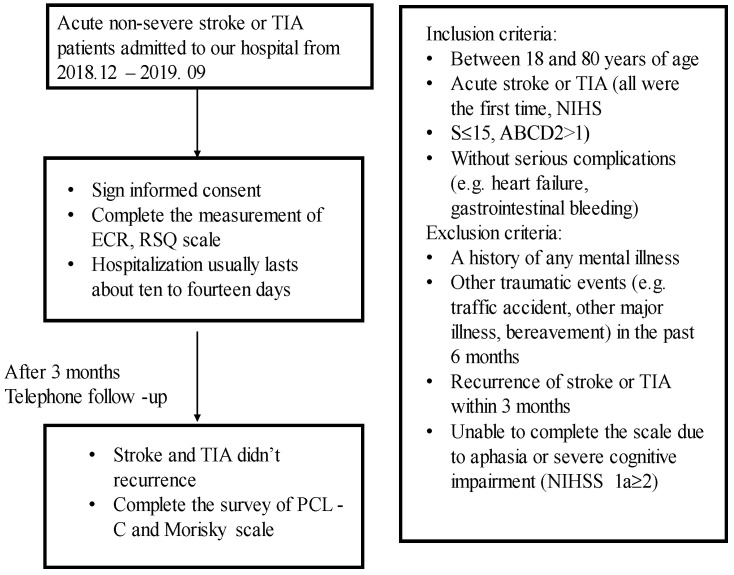
The study flow diagram.

**Figure 2 brainsci-12-01118-f002:**
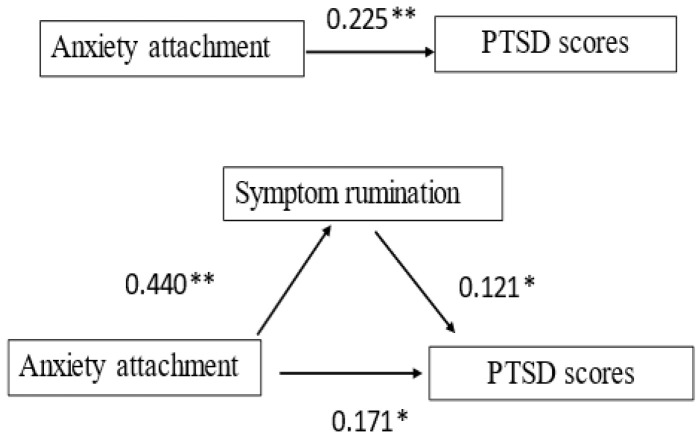
The mediating effect of symptom rumination on attachment anxiety and PTSD scores. ** *p* < 0.01, * *p* < 0.05.

**Figure 3 brainsci-12-01118-f003:**
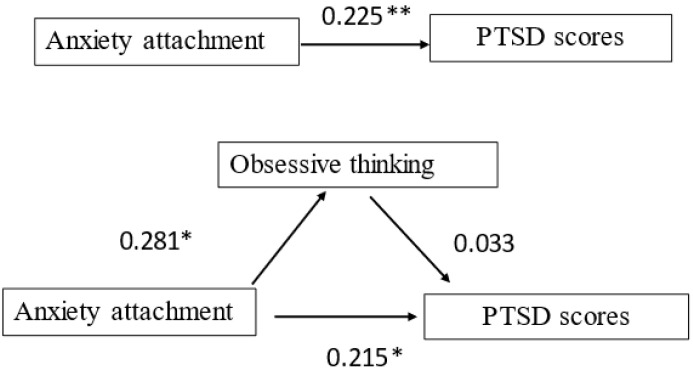
The mediating effect of obsessive thinking on attachment anxiety and PTSD scores. ** *p* < 0.01, * *p* < 0.05.

**Table 1 brainsci-12-01118-t001:** The mean, standard deviation, and correlation coefficient of each measurement index score (*n* = 274).

Variables	1	2	3	4	5	6	7
1. Avoidance attachment	-						
2. Anxiety attachment	0.318 **	-					
3. Symptom rumination	0.173 **	0.440 **	-				
4. Obsessive thinking	0.068	0.305 **	0.725 **	-			
5. Reflective think	0.114	0.281 **	0.632 **	0.670 **	-		
6. PTSD scores	0.070	0.225 **	0.197 **	0.187 **	0.094	-	
7.Medication compliance	−0.080	−0.062	−0.047	−0.050	−0.009	−0.065	-
Mean	3.1888	3.0777	19.1460	9.6168	7.8869	23.9818	4.9407
SD	0.84993	1.17103	5.69884	2.80290	2.51114	5.49089	1.52989

Note. ** *p* < 0.01, PTSD—post-traumatic stress disorder.

**Table 2 brainsci-12-01118-t002:** The effect of attachment anxiety on PTSD scores: the mediating role of symptom rumination.

	PTSD Scores	Symptom Rumination	PTSD Scores
	Model 1	Model 2	Model 3
Predictor variable			
Anxiety attachment	0.225 **	0.440 **	0.171 *
Mediate variable			
Symptom rumination			0.121 *
R^2^	0.051	0.194	0.062
△R^2^	0.047	0.191	0.056
F	14.481 **	65.390 **	9.023 *
△F			5.458 *

Note. ** *p* < 0.01, * *p* < 0.05, PTSD—post-traumatic stress disorder.

**Table 3 brainsci-12-01118-t003:** The effect of attachment anxiety on the PTSD scores: the mediating role of obsessive thinking.

	PTSD Scores	Obsessive Thinking	PTSD Scores
	Model 4	Model 5	Model 6
Predictor variable			
Anxiety attachment	0.225 **	0.281 **	0.215 *
Mediate variable			
Obsessive thinking			0.033
R^2^	0.051	0.079	0.052
△R^2^	0.047	0.075	0.045
F	14.481 **	23.266 **	7.369 **
△F			7.112 *

Note. ** *p* < 0.01, * *p* < 0.05, PTSD—post-traumatic stress disorder.

## Data Availability

The original contributions presented in the study are included in the article.

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
