# Peer review of "Underlying Role of Rumination-Mediated Attachment Style Plays in PTSD after TIA and Stroke"

_brainsci, 2022, doi:10.3390/brainsci12091118_

Round 1

Reviewer 1 Report

Comments and Suggestions for Authors

This is an interesting study. I think it would be useful if your demographic characteristics of the 21 subjects who presented with probable PTSD symptoms is included in the body of the report in a short table. 

 I think the demographics of the 21 subjects presenting with probably PTSD symptoms should be included in a table within the paper. there are limitations to the paper including the fact that attachment styles vary with age. The relatively small sample of 21 probably PTSD subjects is also a limitation and the findings of the study my not be generalizable.

Author Response

Dear reviewer,

      Thank you very much for your thoughtful suggestions. Please see the attachment.

                                                                     Sincerely,

                                                                             Weiping,Wang

Reviewer 2 Report

Comments and Suggestions for Authors

The authors were researching the role of attachment style in mediating PTSD following stroke. The investigation of psychological mechanisms that can prevent PTSD development after stroke (and perhaps be predictive prior to stroke) is an important area.

The authors’ use of English is somewhat shaky, but I would say this requires only moderate revisions. This is less concerning than the fact that language issues may have hindered their understanding of some of the background research. For example, in the Introduction they argue that the prevalence of PTSD following TIA or stroke correlates with a number of factors (according to several authors). But when you go back to these sources, one reports “Older age (OR .93, .90-.95), marriage or partnership (OR .52, .28-.98), and having emotional support (OR .25, .11-.54) were protective against developing PTSD”, while the other says “We found no statistically significant correlation of PTSD with age, gender and marital status”. Thus, it is difficult to ascertain the current state of the literature based on their summary. I would suggest reworking this section to better capture the previous evidence.

A bit more elaboration on the subject demographics would be helpful. For instance, nearly 40% of their sample had only primary school education; here in North America that would be quite remarkable. Is that similar to the population breakdown more generally in Hebei province? More importantly, there is no breakdown of what percentage of each group actually developed PTSD symptoms. While 2/3 of the stroke patients are male, did the PTSD show a similar breakdown? It would be useful to add three extra columns and detail how many of each group were in the no/moderate/high symptom categories.

Their finding, that both obsessive thinking and symptom rumination, appear to mediate the effects of attachment anxiety on PTSD is interesting and important. However, it does seem a bit strange that two mediators (rumination and obsessive thinking) were not combined into a single model. In that case, it may have been possible to see whether one played a more important role than the other, and whether there was any interaction between the two. The authors may want to consider doing this.

The authors’ main conclusion that “When stroke and TIA occur suddenly, patients with attachment anxiety seem to experience symptom rumination, obsessive thinking, trying to cope with illness in an unhealthy way” strikes me as being central, and deserves further discussion. The authors may want to expand on this section. How does this, perhaps, compel them to approach people at high risk for stroke? What is important for them to understand, and is there anything they can change to reduce how negatively it could affect them? Similarly, they should provide more details on treatment implications – while they discuss ways to control rumination, presumably the OCD literature similarly has an abundance of advice on how to reduce obsessive thinking. This would appear to be key to diverting patient risk for PTSD.

Author Response

Dear reviewer,

   Thank you very much for your thoughtful suggestions.Please see the attachment.

                                                                              Sincerely,

                                                                                    Weiping,Wang

Round 2

Reviewer 2 Report

Comments and Suggestions for Authors

I feel that my concerns have been adequately addressed. Thanks to the authors for taking the time to make these changes.

Author Response

Dear reviewer:

        Thank you.Your opinions are very valuable.                                                       

                                                   Yours sincerely

                                                   Weiping Wang